# Flexible and Printed Electrochemical Immunosensor Coated with Oxygen Plasma Treated SWCNTs for Histamine Detection

**DOI:** 10.3390/bios10040035

**Published:** 2020-04-10

**Authors:** Bajramshahe Shkodra, Biresaw Demelash Abera, Giuseppe Cantarella, Ali Douaki, Enrico Avancini, Luisa Petti, Paolo Lugli

**Affiliations:** Faculty of Science and Technology, Free University of Bolzano-Bozen, 39100 Bolzano, Italy; BiresawDemelash.Abera@natec.unibz.it (B.D.A.); Giuseppe.Cantarella@unibz.it (G.C.); Ali.Douaki@natec.unibz.it (A.D.); Enrico.Avancini@unibz.it (E.A.); Paolo.Lugli@unibz.it (P.L.)

**Keywords:** flexible electronics, immunosensor, SWCNTs, biogenic amines, fish

## Abstract

Heterocyclic amine histamine is a well-known foodborne toxicant (mostly linked to “scombroid poisoning”) synthesized from the microbial decarboxylation of amino acid histidine. In this work, we report the fabrication of a flexible screen-printed immunosensor based on a silver electrode coated with single-walled carbon nanotubes (SWCNTs) for the detection of histamine directly in fish samples. Biosensors were realized by first spray depositing SWCNTs on the working electrodes and by subsequently treating them with oxygen plasma to reduce the unwanted effects related to their hydrophobicity. Next, anti-histamine antibodies were directly immobilized on the treated SWCNTs. Histamine was detected using the typical reaction of histamine and histamine-labeled with horseradish peroxidase (HRP) competing to bind with anti-histamine antibodies. The developed immunosensor shows a wide linear detection range from 0.005 to 50 ng/mL for histamine samples, with a coefficient of determination as high as 98.05%. Average recoveries in fish samples were observed from 96.00% to 104.7%. The biosensor also shows good selectivity (less than 3% relative response for cadaverine, putrescine, and tyramine), reproducibility, mechanical and time stability, being a promising analytical tool for the analysis of histamine, as well as of other food hazards.

## 1. Introduction

Every year, millions of cases of food-borne diseases caused by different hazards occur worldwide [1]. Detection and quantification of these hazards in food, as well as protection against spoilage caused by them, is, thus, a key task of great public health importance [2]. Among the most widely known foodborne toxicants, biogenic amines are a class of organic compounds formed through the decarboxylation of amino acids by microorganisms [3]. In particular, histamine, 2-(1H-imidazol-4-yl)ethanamine is a heterocyclic amine, synthesized from the microbial decarboxylation of the amino acid histidine [4]. Ingestion of sufficient quantities of histamine in food—typically in spoiled fish—is usually associated with “scombroid poisoning”, a widely known foodborne intoxication [5]. In the European Union, histamine concentrations higher than 200 mg/kg are considered toxic. In the United States, the Food and Drug Administration imposes a limit of 50 mg/kg [6]. The analysis of histamine in food is, therefore, of great importance not only for its potential risks on human health but also in view of its function as a chemical indicator of food quality and thereby, as an enabler for the valuation of food processing conditions and microbial contamination [1,2,7]. Nowadays, quantification of histamine is performed mainly by chromatography techniques, such as gas chromatography (GC) [8], high-performance liquid chromatography (HPLC) [9], thin-layer chromatography (TLC) [10], capillary isotachophoresis [11], as well as by enzyme-linked immunosorbent assay (ELISA) [12]. The abovementioned techniques are all sensitive and reproducible, but typically require expensive equipment, extensive analysis time, and specifically trained personnel. This is why low-cost, rapid, specific, and sensitive analytical methods for the detection of histamine are strongly needed. In this regard, biosensors offer an attractive analytical tool [7]. Most histamine biosensors reported to date are based on enzymatic reactions using amine oxidases (e.g., [13,14,15]). The specificity of this enzymatic reaction is, however, typically low, as amine oxidase catalyzes histamine oxidation as well as the other amines present in the sample. This cross-reactivity leads, thereby, to a poorly selective sensor. As a viable alternative, immunosensors offer higher specificity due to the specific chemical antibody-antigen interaction [16,17,18,19]. In a recent study, we developed a screen printed immunosensor for histamine detection based on hybrid spray deposited film of poly(3,4-ethyldioxythiophene)poly(styrenesulfunate) (PEDOT:PSS) and single-walled carbon nanotube (SWCNT) films [20]. In this work, we developed an immunosensor based on a direct competitive immunoassay where free histamine competes with horseradish peroxidase (HRP)-conjugated histamine to react directly with immobilized anti-histamine antibodies on the surface of the coated working electrode with oxygen plasma-treated single-walled carbon nanotubes (SWCNTs). To improve the sensitivity, the working electrode was modified by spray deposition of SWCNTs. SWCNT layers were used in this work due to excellent physical/chemical properties, such as large surface area/volume ratios, high electrical conductivity, biocompatibility, mechanical strength, and unique nanoscale interactions [21,22]. To overcome the hydrophobic nature of SWCNTs [23,24]—which challenges biomolecule immobilization, the surface chemical properties of SWCNTs were modified with an oxygen plasma (OP) treatment. OP treatment grafted polar groups onto the surface of SWCNTs and may have allowed higher antibody immobilization and thus, enhanced sensor performance as compared to the previous work [20]. This technique allowed realizing a flexible electrochemical biosensor with a screen-printed three-electrode system based on silver and silver chloride electrodes on flexible polyethylene terephthalate (PET). The demonstrated sensors showed good specificity against other biogenic amines (cadaverine, putrescine and tyramine) thanks to the high selectivity of antibodies as biological receptors, which were immobilized on OP-treated SWCNTs on the working electrode. The so-realized immunosensor was used to determine the concentration of histamine in a range of 0.5 to 50 ng/mL in real fish samples.

## 2. Materials and Methods

### 2.1. Reagents

Histamine, antihistamine antibodies, tyramine, cadaverine, putrescine, phosphate buffer saline (PBS), potassium chloride (KCl), potassium ferricyanide III (K_3_[(Fe(CN)_6_]), potassium ferricyanide II trihydrate (K_4_[(Fe(CN)_6_]x3H_2_O), polyvinyl alcohol (PVA), sodium dodecyl sulfate (SDS) were purchased from Sigma Aldrich. Horseradish peroxidase (HRP) linked with Histamine and 3,3′,5,5′-tetramethylbenzidine (TMB) were purchased from Prodotti Gianni (Italy) as part of a Histamine EIA Kit (EA31). Polyethylene Terephthalate (PET) was purchased from Mylar, USA. Ink-pastes silver ECI 1011 and silver/silver chloride ECI 6038E were obtained from LOCTITE E&C, USA. SWCNTs were acquired from Carbon Solution Ink, CSI, USA. All chemicals used in this work are analytical grade and were used without any further purifications. All the solutions were prepared in the volumetric flask of A category. VWR automatic pipets (2–20 µL ± 1.0—±0.6%, 10–100 µL ± 0.6—±0.5%, 100–1000 µL ± 0.9—±0.6%) were used for the dilutions of the stock solutions.

### 2.2. Sensor Fabrication and Oxygen Plasma Treatment

A typical three-electrode sensor layout was transferred on the PET flexible substrate via screen-printing with an automatic screen-printing machine (Aurel automation S.P.A. C290, Italy) and Ag and AgCl polymeric pastes. In particular, a silver working electrode (WE), a silver counter electrode (CE), and a silver/silver chloride reference electrode (RE) with a total length and width 22 mm and 8 mm, respectively, were prepared (Figure 1). First, WE and CE were printed and cured at 120 °C for 15 min. Subsequently, RE was printed and cured at 120 °C for 15 min, according to [20,25]. To remove any remaining polymers and other impurities from the electrodes [26], the electrodes were then treated in a plasma chamber (Diener electronics, plasma surface technology, Germany) by exposing to oxygen plasma (OP) at a flow rate 6 cm^3^/min, a constant pressure 0.4 mbar for 30 s, and a power of 60 W.

### 2.3. Synthesis, Spray Deposition, and OP Treatment of SWCNTs

A dispersant water solution (SDS 0.5 wt%. in distilled water) was prepared by stirring for 1 h at room temperature. Afterward, 0.05 wt% of SWCNTs was added and dispersed by sonication for 30 min, using a horn sonicator at 50% of the maximum amplitude (Fisher brand Q500). Finally (as previously employed by Falco et al. 2014) [23], the so-dispersed solution was centrifuged at 13,000 rpm for 110 min using a ThermoScientific SL16R centrifuge equipped with an F16-6x Bucket. The prepared SWCNTs dispersion was then spray deposited on the OP-treated WE, using a shadow mask through an automated spray system (Nordson E4 EFD, UK). The sample to nozzle distance was kept constant to 5 cm, whereas the hot plate temperature was set to 70 °C to evaporate the solvent (as reported in Abdelhalim et al. 2013) [27]. After process calibration and optimization through cyclic voltammetry, it was found that 100 spray cycles showed adequate results in terms of current regeneration (see Appendix A). In addition, the surface morphology of the electrodes and the SWCNTs network was characterized by atomic force microscope (AFM). To overcome the hydrophobicity of the SWCNTs [28], a second OP treatment was performed on the full sensor layout after spray deposition of SWCNTs on the WE, using a 6 cm^3^/min flow rate at a constant pressure of 0.4 mbar, and a power of 24 W for 30 s.

### 2.4. Immunosensor Development

The immunosensor realization was based on a direct ELISA principle, where the antihistamine antibodies were placed onto the OP treated SWCNTs-modified WE. A concentration of 10 µg/mL antihistamine antibody (as previously employed by Yang, Zhang, and Chen 2015) [29] in 0.1 M phosphate buffer saline (PBS) pH = 7.4 was used. Seven microliters of the antihistamine antibody was drop casted on the WE, and the sensors were stored overnight at 4 °C, to allow passive adsorption of the antibody onto the OP treated SWCNTs surface. The sensor was then washed with 50 µL of 0.05% Tween 20 in 10 mM PBS buffer, acting as a washing solution [17]. Afterward, 7 µL of polyvinyl alcohol (PVA) (blocking solution) was placed on the electrode surface and incubated at 37 °C for 1 h. PVA is one of the best reagents reported to block any remaining active sites ensuring exclusive specific binding [30,31]. Finally, the sensor was washed and stored at 4 °C until ready to be used. For the sensor calibration, histamine standards were prepared by dissolving the histamine powder in 0.1 M PBS at a concentration of 0.5 µg/mL to prepare a stock solution. Working standard solutions (between 0.005 and 50 ng/mL) were prepared by diluting the stock with 0.1 M PBS buffer. To test the sensor in real sample, fish samples were prepared by diluting the stock solution with the extracted suspension from fish [18]. The immunosensor was developed based on a competitive ELISA. For the competitive reaction, a mixture of 4 µL of the free histamine (diluted on PBS or fish extract) and 4 µL histamine-horseradish peroxidase (HRP) directly from histamine EIA Kit (EA31) without any dilution, was placed onto the WE (Appendix A). The competitive reaction between the free histamine and the histamine-HRP to bind with the anti-histamine antibody was performed for 2 h at 37 °C. The sensor was then washed again and dried. For the electrochemical measurements, the three electrodes were covered with 50 µL of 5 mM 3,3′,5,5′-tetramethylbenzidine (TMB) and 1 mM hydrogen peroxide (H_2_O_2_) in citrate buffer containing 0.1 M KCl as a supporting electrolyte [17]. In this reaction, HRP is oxidized by the H_2_O_2_, and then the oxidized HRP is reduced by oxidizing TMB by losing 2e^−^ [32]. After 15 min, 1 M H_2_SO_4_ was added as a stopping solution and the electrochemical current was measured using a source meter (KEITHLEY 2614B SourceMeter^®^, Tektronix Company, USA), running in chronoamperometry mode, using a custom-made LabVIEW 2017 script with a potential 100 mV, a delay time 0.25 s and a time step of 50 ms. Data collection in the chronoamperometry mode was taken for 5 min.

### 2.5. Electrical and Mechanical Characterization

The morphology of the electrodes and the SWCNT network were analyzed using an atomic force microscope (AFM) (CoreAFM, Nanosurf, Sweden), operating in static force mode. The force setpoint was set to 20 nN using silicon AFM probes (ContAl-G) and a resonant frequency of 13 kHz. AFM picture analysis and the value for the root mean square (Rq) were obtained using the data analysis software Gwydion.

All cyclic voltammetry (CV) and chronoamperometric measurements were performed using a source meter (KEITHLEY 2614B SourceMeter^®^, a Tektronix Company, USA). The CV measurements of electrodes were performed by covering the three electrodes with 50 µL of 1 mM [Fe(CN)_6_]^3−/4−^ containing 0.1 M KCl solution, at a scan rate 100 mV/s. The chronoamperometry measurements were done by adding 50 µL of 5 mM TMB and 1 mM H_2_O_2_ in citrate buffer containing 0.1 M KCl under a fixed potential of −0.1 V vs. the Ag/AgCl electrode [17].

To demonstrate the mechanical flexibility of our sensor, bending tests were performed by dynamically bending the sensor down to a bending radius of 4.4 mm, using a custom-made bending setup.

### 2.6. Preparation of Fish Samples

The preparation of fish samples was based on a green extraction, as described by Yilmaz and Inan [33]. In accordance with this method, the fish was cleaned and cut in small slices, 10 g of fish samples were then finely homogenized using a domestic blender. Five grams of homogenized fish samples were weighed into a test tube and added to 40 mL of distilled water; the mixture was vortexed for 5 min and then sonicated for 20 min in an ultrasonic bath. The resulted mixture then was centrifuged for 20 min at 40,248× *g* (12,000 rpm), while the supernatant then filtered through a Whatman filter. Finally, a second filtration was performed, before using the final suspension to prepare different concentrations of histamine for the sensor testing.

## 3. Results and Discussion

### 3.1. Surface Characterization and Electrochemical Properties of Histamine Immunosensor

AFM imaging was used to characterize the morphology of bare and OP-treated electrodes, as well as to thoroughly analyze the network of SWCNTs. The surface roughness (Rq) of the electrodes before and after OP treatment was measured by AFM. Figure 2a shows the morphology of an untreated silver electrode with Rq of 4.84 µm. After treatment with OP, the surface appears smoother with Rq of 1.08 µm (Figure 2b), possibly because of the removal of the binder polymers and other impurities presented in the silver ink [26]. Figure 2c shows an AFM micrograph of 100 layers of the SWCNTs network treated by OP on a glass substrate with Rq of 57.86 nm. AFM micrographs of untreated SWCNTs showed comparable results in terms of SWCNTs morphology.

The thickness of the silver screen-printed electrode was measured by a non-contact 3D-optical profilometer (ProFilm3D from Filmetrics, Unterhaching, Germany). The 2D profile for the thickness measurement is given in Appendix A, where the thickness was measured in terms of step height. The step height of the silver electrode was 5.38 µm.

The spray-deposited SWCNT layer was treated by OP to modify its surface chemistry and reduce its hydrophobicity. SWCNTs (on OP treated WE) were treated with different OP powers at values of 9, 15, 24, 30, and 39 W for 30 s. To observe the current generation after this step and the difference between the powers applied, CV was performed at a scan rate of 100 mV/s, in 1 mM [Fe(CN)_6_]^3−/4−^ containing 0.1 M KCl solution. As shown in Figure 3a, increasing the OP power from 9 W to 24 W enhanced the oxidation/reduction current peaks reaching a maximum of 1.96 × 10^−2^ A for an OP power of 24 W. This current enhancement can be related to a possible degradation of SDS from SWCNTs network. By further increasing OP power from 24 W to 39 W, the generation of oxidation/reduction current was reduced, potentially due to the chemical etching of SWCNTs at high power (as previously indicated by Ham et al. 2014) [28], as well as due to the increase in defect density on the SWCNTs surface [34].

Besides the higher current generation, the OP treatment leads to the formation of carbonyl and/or carboxylic groups, as reported in the literature [28]. The presence of these groups can improve the immobilization of antibodies on the surface of SWCNTs. Fourier-transform infrared (FTIR) spectroscopy results (Appendix A) indicated the presence of O–H, C–O stretching vibrations rather than carbonyl groups could be the consequence of different initial surface conditions and different OP parameters. The reason for the high noise observed in the region of −0.8 to −0.6 V is unclear. However, this does not affect the interpretation of the result since the oxidation/reduction potential of the [Fe(CN)_6_]^3−/4−^ was far from this low S/N area.

The CVs of the electrodes in different process steps are shown in Figure 3b. After each treatment, the oxidation/reduction currents peaks were enhanced. The values for oxidation current followed this trend: 8.16 × 10^−3^ A for bare electrode, 1.03 × 10^−2^ A for OP treated electrode, 1.50 × 10^−2^ A for spray deposited SWCNTs on OP treated electrode, and 1.96 × 10^−2^ A for OP treated SWCNTs on OP treated electrode. A maximum increase by 2.38 was observed between bare and OP treated SWCNTs on OP treated electrode. We conclude that a substantial (more than twice) increase in current (as compared to the current achieved on bare electrodes) can be obtained by a combination of OP treated electrodes coated with OP-treated SWCNTs. It is clear that the OP treated SWCNTs spray deposited on OP treated electrodes have more favorable electron transfer kinetics.

### 3.2. Histamine Immunosensor Performance

The histamine immunosensor was developed based on the competitive ELISA principle. In the typical competitive ELISA, usually an anti-primary antibody is immobilized first on the surface of the electrode, followed by the immobilization of the second antibody [17]. This method involves two incubation times (two different antibodies) and additional washing and blocking step. For faster detection time and to simplify the immunoassay, we immobilized the anti-histamine antibodies on top of modified WE with the oxygenated SWCNTs directly. The electrochemical signal was obtained as a result of a competitive reaction between histamine and histamine-HRP, as explained in detail in Section 2.4 (Immunosensor development).

The developed immunosensor based on oxygenated SWCNTs and the competitive ELISA was applied to detect histamine at the following concentrations: 0.005, 0.01, 0.05, 0.1, 0.5, 1, 5, 10, and 50 ng/mL in 0.1 M PBS buffer solution. As shown in Figure 4, the electrochemical current produced by the His immunosensor was inversely proportional to the logarithm of the histamine concentration in the sample. A large amount of histamine in the sample caused low current generation as a result of low histamine-HRP bound to the antihistamine antibodies, which led to less HRP present in the sample to catalyze the oxidation of TMB.

The calibration curve (as inset in Figure 4) for histamine immunosensor was realized measuring histamine at concentrations range 0.005 to 50 ng/mL in 0.1 M PBS buffer solution. The measured current was an average of triple measurements at each concentration of histamine, which are included in the calibration curve as a standard deviation of each concentration. A linear response was found between the corresponding current variation (ΔI) and the logarithm concentration of histamine. The found relationship is given by: ΔI[A] = (−7.71 × 10^−6^ ± 1.55 × 10^−7^) logC_Histamine_ [ng/mL] − (2.22 × 10^−5^ ± 4.16 × 10^−7^), with a coefficient of determination 98.05% indicating that the equation is an excellent linear fit for the data. The wide linear range (0.005–50 ng/mL) is attributed to the excellent electrocatalytic properties of the large surface area/volume ratios of the electrodes treated with oxygenated SWCNTs. The limit of detection (LOD) was calculated using Equation (1):
LOD = 3.3 STDEV I_0_/m (1)
where STDEV I_0_ is the standard deviation of the current regeneration for histamine 0 ng/mL and m is the slope which is calculated using Equation (2):
m = (I_1_ − I_0_)/(C_1_ − C_0_)(2)
where I_1_ is the generated current for concentration C_1_ (0.005 ng/mL), and I_0_ is the generate current for the C_0_ blank measurement. LOD was calculated to be 2.48 pg/mL. The performance of our screen-printed immunosensor (linear range 0.005–50 ng/mL and LOD of 2.48 pg/mL) is in good agreement with the immunosensor developed on a conventional glassy carbon electrode (0.001–1 ng/mL and LOD of 0.5 pg/mL) by Yang, Zhang, and Chen 2015 [29].

### 3.3. Selectivity and Reproducibility of Histamine Immunosensor

Histamine is one of the biogenic amines that can be found in food, but other amines may be present as well, so it is very important to test the selectivity of the developed immunosensor. We tested our immunosensor against 5 ng/mL of cadaverine, putrescine, and tyramine, obtaining detected concentrations of 3.01%, 1.20%, and 0.02% of that to 5 ng/mL of histamine, thus demonstrating high selectivity. Besides selectivity, the reproducibility of the immunosensor is also a significant parameter for evaluating its practical application. Here, reproducibility was tested by six measurements at histamine concentrations of 0.01, 0.1, 0.5, 5, and 50 ng/mL, which resulted in relative standard deviations of 2.58%, 3.74%, 3.94%, 6.62%, and 5.06%. Reproducibility strongly depends on the fabrication process, especially for what concerns the homogeneity of the ink [25]. In this work, to achieve a good reproducibility, we combined screen printing of homogeneously dispersed silver and silver chloride electrode inks with a uniform spray coating of SWCNTs layers, followed by solid attachment of antibodies on the oxygenated SWCNT surface.

### 3.4. Flexibility, Regeneration, and Time Stability of Histamine Immunosensor

The use of the sensor for food quality and safety analysis can include mechanical stress. The possible roll-to-roll printing of such flexible electrochemical sensors—to reduce space constraints and the fabrication cost—include mechanical deformation as well. For this reason, high mechanical stability is required, meaning the strength of such devices against mechanical deformation. The flexibility of our sensor was evaluated by a bending test (Figure 5a), where the electrodes were mechanically deformed down to a radius of 4.4 mm. As shown in Figure 5b, the immunosensor response was recorded after 0, 100, 250, 500, and 1000 bending cycles for a histamine concentration of 5 ng/mL. The current generation was well maintained after such repetitive bending cycles, demonstrating the stability of the sensors counter to mechanical deformations. However, the increased standard deviation after the bending cycles (±0.23 µA for 0 bending cycles to ±0.74 µA for 1000 bending cycles) indicates an effect of mechanical deformation on the measurement precision. Future work will focus on substrate and ink modification, as well as improving the antibodies on the WE, to reduce the influence of deformation on the sensor performance.

A regeneration of the immunosensor, together with the bending test, was performed (Appendix A). For the regeneration of the immunosensor, a 10% Tween-20 in 0.1 M PBS [35] was used to dissociate the bond between histamine and the ant-histamine antibody. After the second regeneration and 250 bending cycles, the current generation was 94.66% with a standard deviation of ± 0.78 µA, indicating that the immunosensor can be reused up to three times.

To examine the shelf life of our immunosensor, the sensor stability over time was also tested. In this experiment, the sensors were fabricated and stored at 4 °C and tested for the detection of 5 ng/mL histamine every 2 days for one month. After 12 days, the current generation was 95.29% of its initial value (as displayed in Appendix A). Starting from day 14, the current generation dropped by 39.96%, most probably due to loss of the sensitivity of the antihistamine antibodies. As such, an abrupt response is somehow unexpected, possible causes of this are mishandling or other unexpected changes in the test. Nevertheless, 12 days-stability is confirmed.

### 3.5. Histamine Detection in Fish Samples

To assess the matrix effect on the sensor response, the developed immunosensor was used to determine histamine concentration in fish samples. The standard addition method was used to prepare 0.5, 1, 5, and 50 ng/mL (n = 3) in the extracted suspension from fish.

The current values were similar to those found in the buffer solution from the calibration curve, indicating that the matrix effect of the real samples was negligible on the immunosensor response. Average recoveries (Table 1) were observed from 96.00% to 104.7%, with the coefficient of variance ranging from 3.20% to 8.5%.

## 4. Conclusions

We demonstrated flexible histamine immunosensors printed on PET substrate. The design of the immunosensor was based on a competitive ELISA principle, where immobilization of anti-histamine antibodies on OP treated SWCNTs led to specific and sensitive histamine detection with 2.48 pg/mL as LOD. This study shows that the proposed immunosensor is a promising analytical tool for fast and low-cost analysis for histamine—one of the most potent food hazards, as confirmed by thoughtful and detailed experiments performed. Moreover, the realized His sensor has a shelf life of 12 days, and it maintains the functionality after repetitive mechanical deformations. Nevertheless, a decrease in precision after increased bending cycles (1000 cycles) was observed. The developed immunosensor is a promising analytical tool for low-cost, rapid, specific, and sensitive histamine detection. Future work will focus on substrate and ink optimization, as well as on improving the attachment of antibodies on the SWCNTs to improve mechanical properties and, therefore, full sensor reliability in multiple applications.

## Figures and Tables

**Figure 1 biosensors-10-00035-f001:**
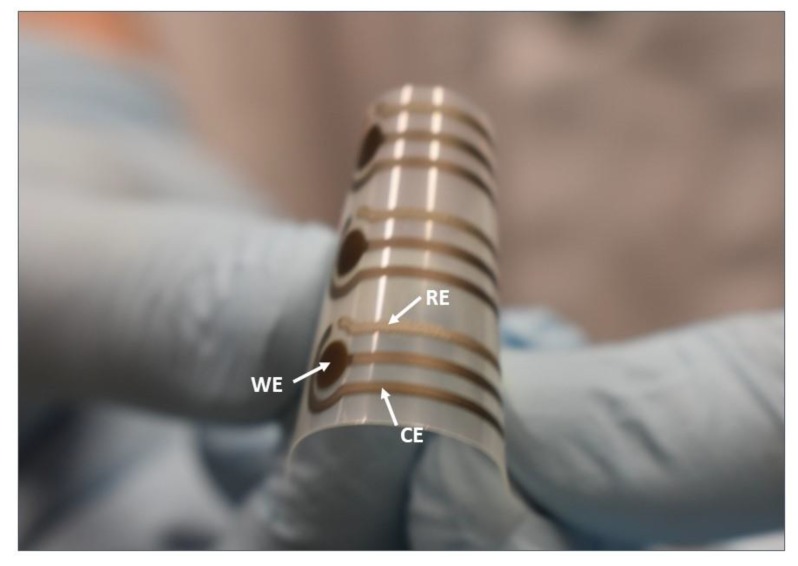
The three-electrode (silver working electrode (WE) and counter electrode (CE), silver/silver chloride reference electrode (RE)) sensor layout screen-printed on flexible polyethylene terephthalate (PET).

**Figure 2 biosensors-10-00035-f002:**
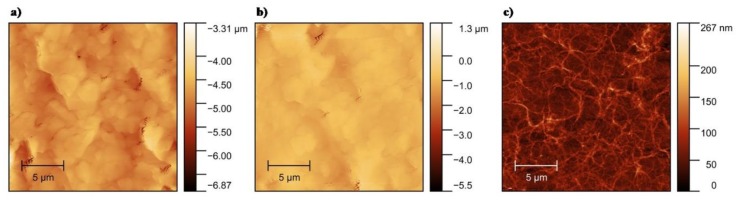
Atomic force microscope (AFM) micrograph of (**a**) printed silver electrode, (**b**) printed silver electrode treated with oxygen plasma (OP), (**c**) OP treated single-walled carbon nanotubes (SWCNTs) on glass.

**Figure 3 biosensors-10-00035-f003:**
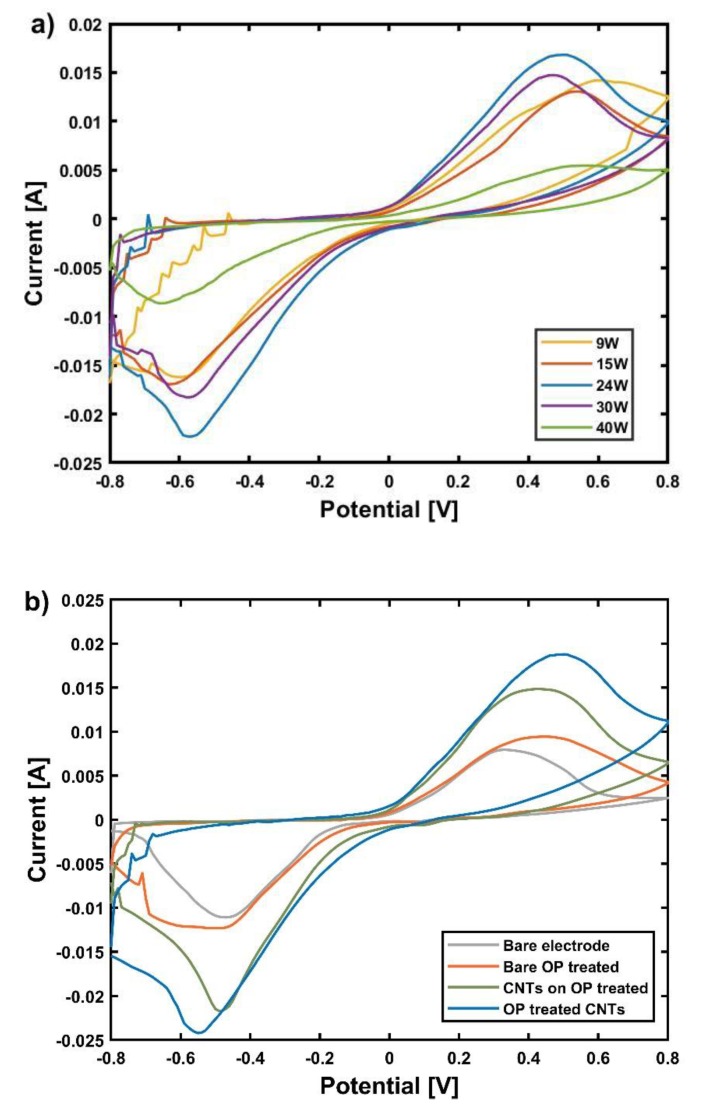
Cyclic voltammograms at a scan rate of 100 mV/s, in 1 mM [Fe(CN)_6_]^3−/4−^ containing 0.1 M KCl solution: (**a**) of OP treated SWCNTs with different OP power, (**b**) for bare, OP treated electrode, spray deposited SWCNTs on OP treated electrode, and OP treated SWCNTs on OP treated electrode.

**Figure 4 biosensors-10-00035-f004:**
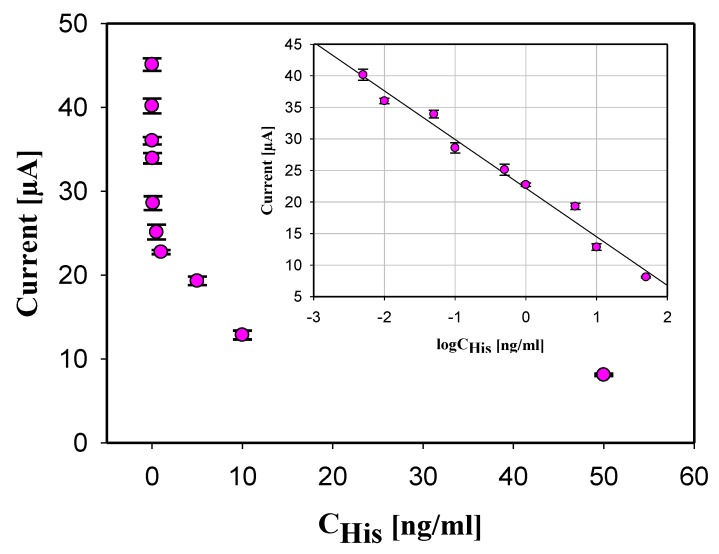
Measured current generation vs. His concentration in the sample. Inserted calibration curve of the histamine immunosensor. Error bars shown as a triple of the standard deviation (n = 3).

**Figure 5 biosensors-10-00035-f005:**
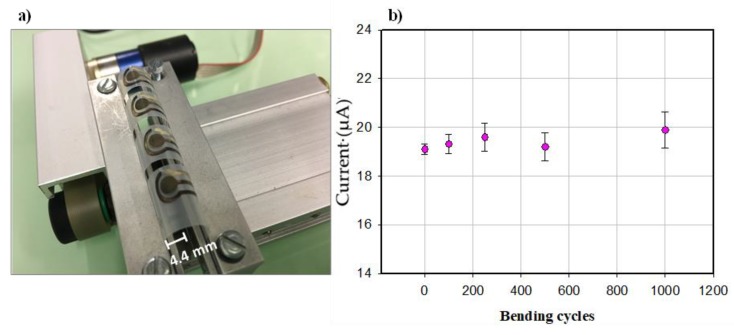
(**a**) Bent sensors with the radius of 4.4 mm and (**b**) Flexibility test for histamine immunosensor done after bending the sensors. Error bars reported as a triple of the standard deviation (n = 3).

**Table 1 biosensors-10-00035-t001:** Recovery test of histamine in fish samples (ng/mL).

Amount of Histamine Added (ng/mL)	0	0.5	1	5	50
Amount of histamine detected	-	0.491 ± 0.032	1.047 ± 0.036	5.061 ± 0.039	48.0 ± 0.085
Recovery %	-	98.0 ± 3.2	104.7 ± 3.6	101.2 ± 3.9	96.0 ± 8.5

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
