# Peer review of "Flexible and Printed Electrochemical Immunosensor Coated with Oxygen Plasma Treated SWCNTs for Histamine Detection"

_biosensors, 2020, doi:10.3390/bios10040035_

Round 1
Reviewer 1 Report
The authors developed and characterize a novel printed electrochemical immunosensor for histamine. This immunosensor showed excellent performance in its sensitivity and selectivity. More detailed and precise information of this sensor should be useful for the readers.
Although immunoglobulin is generally regarded as a relatively stable protein, a potential to bind to its antigen could be attenuated when exposed to lower pH or dried. If possible, the authors should confirm the functional stability of the antibody used in this study using another way of evaluation, such as conventional competitive ELISA.
Is it possible to regenerate and reuse this immunosensor? I suppose that the final acidic condition should irreversibly damage the antibody.
Because "His" generally means an amino acid, histidine, it should be replaced with another abbreviation.
The concentration of His-HRP is not described.
Line 146: Centrifugal force should be described using x g, not rpm, because it depends on the rotor size.
Line 240-243: This statement is not based on the experimental evidence. It might be mere speculation.
Reviewer 2 Report
Biosensors-768764
“Flexible and Printed Electrochemical Immunosensor Coated with Oxygen Plasma treated SWCNTs for Histamine Detection” by B. Shkodra et al.
In this paper, the authors developed flexible and printed electrochemical immunosensor fo histamine detection. It must be excellent devise. But I have several concerns.
Major comments
- The authors used the antibodies to histamine was those used by Yang, Zhang, and Chen 2015. Therefore, the authors should compared to their sensor and the discuss the merit of the present devise.
- Is this sensor only one use? How long it will take to get the result?
- The selectivity was examined by using the same concentration of cadaverine, putrescine and tyramine. How about histidine, it might be existed higher concentration that histamine.
- What is the real histamine content inspoiled fish? This information might be useful.
- Usually, how is the histamine content in the spoiled fish detected? The authors should comment the merit of this devise.
Reviewer 3 Report
The article “Flexible and Printed Electrochemical Immunosensor Coated with Oxygen Plasma treated SWCNTs for Histamine Detection” presents a flexible sensor fabricated screen printing silver electrodes on PET and spray coating SWCNT on top of the silver to increase the sensor electrical properties. Oxigen plasma treatments are performed to increase the performance further. Overall the work is well conducted and described. Good histamine detection with high selectivity is achieved. Minor revisions are suggested below.
Comments:
- The authors should present the work on reference 24 in the introduction (that looks like closer prior art) and explain what the main difference with the presented work is.
- Why the authors sprayed SWCNT and not multi-walled carbon nanotubes (MWCNT) or graphene nanoplatelets (GNP) or carbon nanofibers (CNF)? SWCNTs are more expensive. Do the authors expect to have similar performance employing MWCNT or CNF that have a structure similar to SMCNT? What do they expect to obtain with GNP? Could they comment on this?
- What was the thickness of the Silver electrodes? What was the thickness of the SWCNT layer on top of the silver electrodes? Could the authors provide a cross-sectional SEM image of the sensors?
